REGISTERED REPORT PROTOCOL

# Pertussis immunisation in infancy and atopic outcomes: A protocol for a population-based cohort study using linked administrative data

**Gladymar Pérez Chacón**[1,2], **Parveen Fathima**[1], **Mark Jones**[3], **Rosanne Barnes**[1], **Peter C. Richmond**[1,4], **Heather F. Gidding**[5,6,7], **Hannah C. Moore**[1,2], **Thomas L. Snelling**[1,2,3]*

1 Wesfarmers Centre of Vaccines and Infectious Diseases, Telethon Kids Institute, University of Western Australia, Perth, WA, Australia, 2 Faculty of Health Science, Curtin School of Population Health, Curtin University, Bentley, WA, Australia, 3 Faculty of Medicine and Health, Health and Clinical Analytics Lab, Sydney School of Public Health, The University of Sydney, Sydney, NSW, Australia, 4 Division of Paediatrics, University of Western Australia, Perth, WA, Australia, 5 Northern Clinical School, The University of Sydney, Sydney, NSW, Australia, 6 Women and Babies Health Research, Kolling Institute, Northern Sydney Local Health District, Sydney, NSW, Australia, 7 National Centre for Immunisation Research and Surveillance of Vaccine Preventable Diseases, The Children's Hospital at Westmead, Sydney, NSW, Australia

* tom.snelling@sydney.edu.au

This is a Registered Report and may have an associated publication; please check the article page on the journal site for any related articles.

## Abstract

### Introduction

The burden of IgE-mediated food allergy in Australian born children is reported to be among the highest globally. This illness shares risk factors and frequently coexists with asthma, one of the most common noncommunicable diseases of childhood. Findings from a case-control study suggest that compared to immunisation with acellular pertussis vaccine, early priming of infants with whole-cell pertussis vaccine may be associated with a lower risk of subsequent IgE-mediated food allergy. If whole-cell vaccination is protective of food allergy and other atopic diseases, especially if protective against childhood asthma, the population-level effects could justify its preferential recommendation. However, the potential beneficial effects of whole-cell pertussis vaccination for the prevention of atopic diseases at a population-scale are yet to be investigated.

### Methods and analysis

Analyses of population-based record linkage data will be undertaken to compare the rates of admissions to hospital for asthma in children aged between 5 and 15 years old, who were born in Western Australia (WA) or New South Wales (NSW) between 1997 and 1999 (329,831) when pertussis immunisation in Australia transitioned from whole-cell to acellular only schedules. In the primary analysis we will estimate hazard ratios and 95% confidence intervals for the time-to-first-event (hospital admissions as above) using Cox proportional hazard models in recipients of a first dose of whole-cell versus acellular pertussis-containing vaccine before 112 days old (~4 months of age). Similarly, we will also fit time-to-recurrent events analyses using Andersen-Gill models, and robust variance estimates to account for potential within-child dependence. Hospitalisations for all-cause anaphylaxis, food

**Data Availability Statement:** Authors are not permitted to share individual level data from this study. Data can be requested and accessed through the relevant data custodians and data linkage branches in the states of New South Wales (NSW Centre for Health Record Linkage (https://www.cherel.org.au/) and Western Australia (WA Data Linkage (https://www.datalinkage-wa.org.au/).).

**Funding:** This study was funded by the Population Health Research Network Proof of Concept Project (www.phrn.org.au/), a capability of the Commonwealth Government Collaborative Research Infrastructure Strategy and Education Investment Fund Super Science Initiative, the Australian National Health and Medical Research Council (NHMRC project grant GNT1082342, chief investigator HFG; www.nhmrc.gov.au/) and the Wesfarmers Centre of Vaccines and Infectious Diseases seed funding grant (round 1 - 2018, GPC, HCM, HFG, TLS; infectiousdiseases.telethonkids.org.au/). HFG and HCM are funded by the Australian National Health and Medical Research Council fellowships (www.nhmrc.gov.au/). TS is supported by a Medical Research Future Fund Investigator Grant (MRF1195153; www.health.gov.au/initiatives-and-programs/medical-research-future-fund). GPC is funded by the Australian Department of Education and Training Endeavour Scholarship (www.dese.gov.au/endeavour-leadership-program), Wesfarmers Centre of Vaccine and Infectious Diseases at the Telethon Kids Institute, top-up scholarship (infectiousdiseases.telethonkids.org.au/), and Forrest Research Foundation supplementary scholarship (https://www.forrestresearch.org.au/). The funders had and will not have a role in study design, data collection and analysis, decision to publish, or preparation of the manuscript.

**Competing interests:** I have read the journal's policy and the authors of this manuscript have the following competing interests: PCR reports Institutional funding from a previous grant from GlaxoSmithKline and has served on pertussis scientific advisory boards for GlaxoSmithKline and Sanofi with no personal remuneration that are outside the scope of this publication. The other authors have declared that no competing interests exist.

anaphylaxis, venom, all-cause urticaria and atopic dermatitis will also be examined in children who received at least one dose of pertussis-containing vaccine by the time of the cohort entry, using analogous statistical methods. Presentations to the emergency departments will be assessed separately using the same statistical approach.

## Introduction

Conventional whole-cell pertussis vaccines (wP) were developed between the 1920s and 1940s [1], with subsequent widespread use in combination with diphtheria and tetanus toxoids as a three-dose primary schedule. The reactogenicity of these formulations is thought to be largely driven by the presence of endotoxin, and may manifest as self-limited injection site reactions, fever, irritability and other systemic symptoms [2,3].

The implementation of wP-based primary series in the past century successfully contributed to the near elimination of pertussis in countries like Australia which achieved high immunisation coverage [4]. From 1974, improved access to wP formulations through the Expanded Programme of Immunization has decreased vaccination inequalities in the developing world and the global burden of pertussis disease in children younger than 5 years old [5]. Nevertheless, geographical, social factors and weak health systems still drive vaccine inequity within low and middle income countries, substantially affecting the coverage of the first and third dose of pertussis vaccine primary series, and pertussis-related deaths [5,6]. Development of acellular subunit vaccines (aP) in the 1980s with substantially reduced endotoxin content and improved reactogenicity profile [7], prompted the switchover to aP primary immunisation in most high-income countries over the past 25 years.

In Australia, the transition from wP to exclusive use of aP vaccine formulations for the primary series occurred during the period of 1997 to 1999 [4]. During this period, existing batches of wP were used and fully depleted by community vaccine providers, before being replenished with batches of aP vaccine. This process occurred in a more-or-less stochastic fashion through the jurisdictional immunisation programs. Therefore, the receipt of wP or aP on the day of the first vaccination was not driven by the individual preference of carers or providers, but by which vaccine happened to be available at the primary care services and public immunisation clinics.

The period of switchover from wP to aP in Australia also overlapped with the change from the ninth to the tenth revision of the International Classification of Diseases (ICD) coding scheme. The first edition of ICD-10 AM (AM: Australian Modification) was introduced in New South Wales (NSW) in July 1998, and in Western Australia (WA) in July 1999 [8].

An ecological association has been reported between the transition from wP to aP vaccine in Australia, and an increase in hospital admissions for food-induced anaphylaxis (ICD-9 CM/ICD-10 AM: 995.6/T78) in infants under 12 months old [9]. In the two decades following the replacement of wP, admissions coded as food-associated anaphylaxis increased from 7.3 per 100,000 population in children under 5 years old in 1998/1999 to 30.3 per 100,000 population in 2011/2012 and from 1.7 per 100,000 children 5–14 years old in 2005/2006, to 12.1 per 100,000 children in 2011/2012 [10].

In Australia the prevalence of self-reported "current asthma" in children under 15 years old has declined from 13.5% in 2001 to 9.9% in 2007/2008 [11]. While the rates of all-cause hospitalisation remained stable, paediatric admissions to hospital coded as asthma decreased nationwide by 33% from 1998/1999 to 2010/2011 [12]. Optimisation in the management of

acute exacerbations by carers, modifications in admission practices or in the severity of the presentations have been proposed as potential explanations for this trend, yet it remains uncertain the role played by these factors over this period [12]. Admissions to hospital are more likely to occur in those with more severe symptoms, or in those who did not respond well to the initial management in the emergency department. On the other hand, visits to the emergency not only reflect disease severity, but in some circumstances, socio-demographic disparities that negatively impact the access to primary care.

Diagnosis and coding of anaphylaxis in WA improved after 2012, as a result of the implementation of a new guideline for the management of anaphylaxis, and an intensified allergy training program [13]. While this may have improved the ascertainment of cases in this period, it is likely there was also a true rise in the presentations for anaphylaxis in children predominantly driven by tree nut allergy [13].

In the UK, the replacement of the wP-based primary series commenced in September 2004 [14]. Despite a reported increase in hospitalisations for food anaphylaxis between 1998 and 2018 [15]. in a birth cohort of children born from 1998 to 2010, no association was found between the type of pertussis vaccine administered and admissions coded as anaphylactic shock (T78.0) by 12 months old (In text: In an email from Professor Liz Miller (Liz.Miller@phe.gov.uk) in May 2013).

From 2011, the National Institute for Health and Care Excellence in the UK has recommended that children under 16 years old with anaphylaxis should be admitted to hospital [15]. An immediate increase in hospitalisations was observed and hospitalisations for anaphylaxis have continued to increase over the last 5 years [15].

Previous studies have examined the relationship of priming with pertussis-containing vaccines on the development of atopic outcomes in infancy, but had limited power to detect a modest but clinically important difference. A randomised controlled trial of Swedish children born early 1990s found no difference in the cumulative incidence of atopic diseases at 2.5 years old, nor in the frequency of atopic dermatitis or asthma among children vaccinated with wP, aP or diphtheria and tetanus toxoid (DT) vaccines [16]. Similarly, a retrospective cohort study of children born in the Isle of Wight (UK) between September 2001 and August 2002 (a period of shortage of wP) did not show an association either between the type of pertussis-containing vaccine received as a first dose, or the type of pertussis immunization schedule (i.e. wP-only doses versus at least one dose of aP in fully-vaccinated infants for pertussis antigens), and IgE-mediated food allergy, atopic dermatitis and asthma during a 10-year period of follow-up [17]. A retrospective analysis of data from a study of Australian children born between 1997 and 2000 found no association between vaccine type and risk of asthma or atopic dermatitis [18].

Conversely, a case-control study of Australian infants born between 1997 and 1999, the period of changeover to aP-based schedules, reported a lower likelihood of IgE-mediated food allergy in infants primed with a first dose of wP, compared to those that received a first dose of aP [odds ratio: 0.77; 95% confidence interval (CI), 0.62 to 0.95] [19]. The biological plausibility of this finding rests on the differential T-cell polarisation elicited by each type of vaccine. Whereas early priming with aP induces a T helper 2 ($Th_2$)-skewed immunophenotype, wP skews T-cell specific memory responses towards $Th_1/Th_{17}$. This wP-driven immune bias is hypothesised to overcome the physiologic $Th_2$ polarisation observed in infants, promoting a more tolerogenic immune environment in those with predisposition to atopy [19].

The above 'natural experiment' provided by the programmatic change of pertussis immunisation schedule in Australia, suggests these *in vitro* observations may have clinical consequences. Nonetheless, it remains unproven that a first or subsequent doses of wP in the primary series have a material influence on the subsequent risk of IgE-mediated food allergy, asthma, and or other atopic diseases.

While the current evidence supports the early introduction of peanut and egg into the infants' diet to prevent peanut and egg allergy, respectively [20], the developmental trajectories of asthma are poorly understood and thus, the development of targeted primary prevention strategies is stalled [21].

Since the implementation of randomised controlled trials in settings where wP has been phased-out would require long-term follow-up of large numbers, observational studies using population-based data represent a potentially valuable source of information. In this paper we describe the methods to compare a birth cohort of children vaccinated with a first dose of wP versus aP with respect to the time to a first or recurrent tertiary care encounter for acute exacerbations of asthma and other atopic diseases.

## Methods

### Study design

The analysis plan described in this protocol is part of a larger project, that aimed to assess Australia's National Immunisation Program using probabilistic linked data for 1.95 million children born in WA or NSW between 1996 and 2012, encompassing perinatal, birth registries and hospital data, as well as Commonwealth data collections from the Australian Immunisation Register (AIR)—known as Australian Childhood Immunisation Register or ACIR before 2016—and National Death Index. The details of the record linkage procedures have been published elsewhere [22,23].

The cohort for this study will be restricted to all live births in WA or NSW between 1 January 1997 and 31 December 1999. These will be sourced from state birth registries and perinatal data collections (WA Midwives' Notification System and NSW Perinatal Data Collection) using the above-described linked dataset. The assembled information includes the parent and child's demographics, maternal obstetric and medical history, labour and delivery details, hospital admissions and presentations to the emergency departments. Hospitalisation data are linked for analysis for the period of January 1997 to December 2013 (WA), and July 2001 to December 2013 (NSW). Presentations to the emergency departments are available from all hospitals in WA from January 2002 to December 2013, and from most public hospitals located in the metropolitan areas of NSW from January 2005 to December 2013 [22].

### Variables

**Exposure.** We defined the exposure according to the child's immunisation status.

For the primary analysis, the exposure of interest will be the first dose of pertussis-containing vaccine (wP or aP), irrespective of the type of vaccine given for any subsequent doses.

For the secondary analyses, we will consider the following exposures:

1. Secondary analysis 1: at least one dose of wP versus aP-only pertussis immunisation, in recipients of a three-dose priming schedule (i.e. excluding children with incomplete schedules).

2. Secondary analysis 2: wP-only versus aP-only pertussis immunisation in recipients of a three-dose priming schedule (i.e. excluding children with incomplete or mixed wP/aP schedules).

During the study period, a single wP formulation was available comprising wP in combination with diphtheria and tetanus toxoid (DTPw, Triple Antigen, CSL, Parkville, Australia), whereas aP vaccines were provided as combination vaccines (Infanrix, SmithKline Beecham

and Tripacel, CSL Vaccines, Connaught Laboratories, Canada) comprising DTaP with or without hepatitis B vaccine antigen [24].

The exposure variables will be identified in the AIR dataset, which captures immunisation data from all children enrolled in Medicare, Australia's universal health insurance scheme [22]. Each dose of the primary course of pertussis vaccines represents a vaccination record. For the purpose of this study, each dose is defined by a single date, dose number, type and brand of pertussis vaccines. Any duplicate immunisation records (i.e. same immunisation date and vaccine dose) have been previously excluded.

Consistent with the aforementioned case-control study [19], children will be excluded from the cohort if a first dose of pertussis-containing vaccine was not recorded, or recorded as given before 39 days or after 111 days of age (~ 4 months old).

**Outcomes.** Admissions to hospital for asthma, all-cause anaphylaxis, food anaphylaxis, anaphylactic reaction to *Hymenoptera* stings (hereafter referred as venom), all-cause urticaria, atopic dermatitis and 'allergy' will be ascertained from the WA Hospital Morbidity Data Collection and the NSW Admitted Patient Data Collection.

These datasets contain the dates of admission and separation (i.e. discharge, transfer or death), the primary diagnosis and codes for diagnostic procedures performed during hospital stay [22]. In addition, up to 20 secondary diagnoses for children born in WA, and up to 50 secondary diagnoses for children born in NSW are also provided [22]. Clinical diagnoses were coded using the ICD system (ICD-9 CM and ICD-10 AM) [22]. These diagnostic codes are listed in Table 1.

For the purpose of this study, we will only include for analysis the primary diagnosis.

*Hospitalisations for asthma after 5 years old.* Every eligible child at time zero (i.e. alive at the start of follow-up, and vaccinated with a first dose of wP or aP between 39 and 111 days old) will be included in the cohort. Where the outcome of interest is time-to-first admission to hospital for asthma after 5 years old, time zero will be the date of the fifth birthday; hospitalisations for asthma before 5 years old will be disregarded. The same rules will be applied to the time-to-recurrent event analyses.

The reasons for only including hospitalisations for asthma after 5 years of age are twofold.

Firstly, the molecular phenotypes of children with acute exacerbations of wheezing differ according to the expression of type I interferon signatures [25]. The latter have been found to be upregulated in children with wheezing episodes triggered by respiratory virus infection, but

**Table 1. ICD-9 CM and ICD-10 AM diagnostic codes of interest.**

| Outcome | ICD-9 CM codes | ICD-10 AM codes |
|---|---|---|
| Asthma | 493, 493.0, 493.00, 493.01, 493.02, 493.1, 493.10, 493.11, 493.12, 493.82, 493.2, 493.8, 493.9, 493.90, 493.91, 493.92 | J45, J45.0, J45.1, J45.9, J45.8, J46 |
| All-cause anaphylaxis | 995.0, 995.6, 995.60, 995.61, 995.62, 995.63, 995.64, 995.65, 995.66, 995.67, 995.69, 995.4, 989.5 | T78.2, T78.0, T63.4, T88.6, T88.2 |
| Food anaphylaxis | 995.6, 995.61, 995.62, 995.63, 995.64, 995.65, 995.66, 995.67, 995.69 | T78.0 |
| Venom | 989.5 | T63.4 |
| Urticaria | 708, 708.0, 708.1, 708.2, 708.3, 708.4, 708.5, 708.8, 708.9 | L50, L50.0, L50.1, L50.2, L50.3, L50.4, L50.5, L50.8, L50.6, L50.9 |
| Atopic dermatitis | 691.8 | L20.0, L20.8, L20.9 |
| Allergy | 995.3 | T78.4 |
| Combined injury, trauma and poisoning | 800 to 999, excluding 995, 978, 979 | S00 to T98 excluding, T78.0, T78.2, T78.3, and T78.4 |

in contrast, respiratory virus negative exacerbations are characterised by the upregulation of $Th_2$ associated pathways, downregulation of interferon gamma and increased likelihood of admission to hospital [25]. Unlike asthma in older children, a large proportion of wheezing episodes in preschool aged children are thought to be non-IgE mediated inflammation triggered by respiratory virus infection rather than $Th_2$–mediated inflammation and airway hyperresponsiveness, and are unlikely to be prevented by wP priming.

Second, hospitalisation data for children born in NSW are only available from July 2001 and therefore, the follow-up period between the first dose of wP or aP and this time point is not available.

Admissions to hospital with asthma-related ICD codes (primary diagnosis) occurring within 14 days will be considered the same episode. Those occurring after 14 days from the previous admission will be defined as recurrent episodes.

*Hospitalisations for other atopic outcomes*. For atopic outcomes other than asthma, children born in WA will enter the cohort at 4 months old (time zero), and those born in NSW will enter the cohort at 5 years old.

Hospitalisation for the same atopic outcome occurring within 14 days will be considered the same episode of illness. Those occurring after 14 days from the previous admission will be defined as a recurrent event.

*Emergency department presentations*. For children born in WA, presentations to the emergency department for asthma and other atopic outcomes will be ascertained from the WA Emergency Department Data Collection [22], using a diagnostic hierarchy rule based on data availability, where the most specific diagnostic category is chosen over the less specific [26]. According to this rule, outcome data will be determined from the following sources: diagnosis (ICD code), symptom code, diagnosis at discharge, presenting complaint and major diagnosis category. These data are available from January 2002 to December 2013 and therefore, every model will be fitted using the date of the fifth birthday as the entry date to the cohort.

For children born in NSW, visits to the emergency department for the outcomes of interest will be sourced from the NSW Emergency Department Data Collection, which provides the principal diagnosis. These data are available from January 2005 to December 2013 and thus, time zero will be set at the age of the eight birthday. The diagnoses were recorded with either of ICD 9-CM, ICD-10-AM or SNOMED CT systems.

Presentations to the emergency department for the same atopic outcome occurring within 14 days will be considered the same episode. Recurrent presentations for the same atopic outcome will be defined as those occurring after 14 days from the previous admission.

## Confounders and explanatory variables

Maternal and infant characteristics as recorded on the birth registries and perinatal data collections, will be incorporated in the models as potential confounders of vaccine effects on atopic disease (i.e. as common causes of the exposure and the outcomes of interest), or as prognostically important covariates. Both will be selected *a priori* based on causal considerations and the *modified disjunctive cause criterion* proposed by VanderWeele [27]. aided by graphical models of asthma generated in GeNIe modeler [28], and other atopic diseases (see Estcourt et al [19]). Briefly, these will be restricted to:

- common causes of the exposure and outcome;

- the proxies of unmeasured confounders; and

- those related to the outcome, but not to the exposure.

The proposed set was measured before the administration of the first dose of wP/aP. Collider stratification bias will be avoided by not including variables in the causal pathway from the exposure to the outcome (i.e. those that occur post exposure).

## Statistical analysis plan

The study population characteristics will be described by state of birth and presented using appropriate summary statistics (means and standard deviations for symmetric continuous distributions, medians and interquartile range for asymmetric distributions, and frequency and proportion for categorical and binary variables). Through the cohort flowchart we will summarise the number and proportion of children that met each of the exclusion criteria. We will also provide a summary of the variables listed in Table 2, overall and by type of vaccine received for dose one.

The Index of Relative Socio-economic Advantage and Disadvantage is a general socio-economic index derived by the Australian Bureau of Statistics, which includes 21 different measurements that reflect income, internet connection, occupation and education [29]. The Accessibility/Remoteness Index of Australia is a national measure of geographic remoteness and access to services for populated localities throughout Australia, based on road distance [30]. In our analyses, each of these indices will be determined according to the residential address of the mother at the time of their child's birth.

**Table 2. Variables that will be included in the description of the cohort.**

| Variable | Classification | Description |
|---|---|---|
| Child characteristics | | |
| Sex[a] | Dichotomous | Female or male |
| Gestational age at delivery[a] | An integer measured in weeks | |
| Method of delivery[a] | Categorical | Vaginal, caesarean or instrumental |
| Apgar score (5 minutes) | Categorical | < 7, 7, 8, 9, 10 |
| Birthweight | An integer measured in grams | |
| Aboriginal status[a] | Dichotomous | Aboriginal and/or Torres Strait Islander or non-Aboriginal |
| Season of birth[a] | Categorical | Summer, autumn, winter, spring |
| Age at the first dose of pertussis-containing vaccine[b] | An integer measured in days | |
| Age at first admission to hospital for the above-specified outcomes | An integer measured in months or years | |
| Number of recurrent events for each outcome of interest | An integer | |
| Parental factors | | |
| Maternal age | An integer measured in years | |
| Number of previous pregnancies greater than 20 weeks gestation[a] | Categorical | 0, 1, 2 ≥ 3 |
| Maternal smoking during pregnancy[a] | Dichotomous | Yes or no |
| Mother born overseas | Dichotomous | Yes or no |
| Paternal age | An integer measured in years | |
| Index of Relative Socio-economic Advantage and Disadvantage[a] | Categorical | 0 to 10% or most disadvantaged, 11 to 25%, 26 to 75%, and 91 to 100% or least disadvantaged |
| Accessibility/Remoteness Index of Australia[a] | Categorical | Major cities, inner regional, outer regional, remote and very remote |

[a]These variables will be included in the models.

[b]This variable will be included in the subgroup analysis.

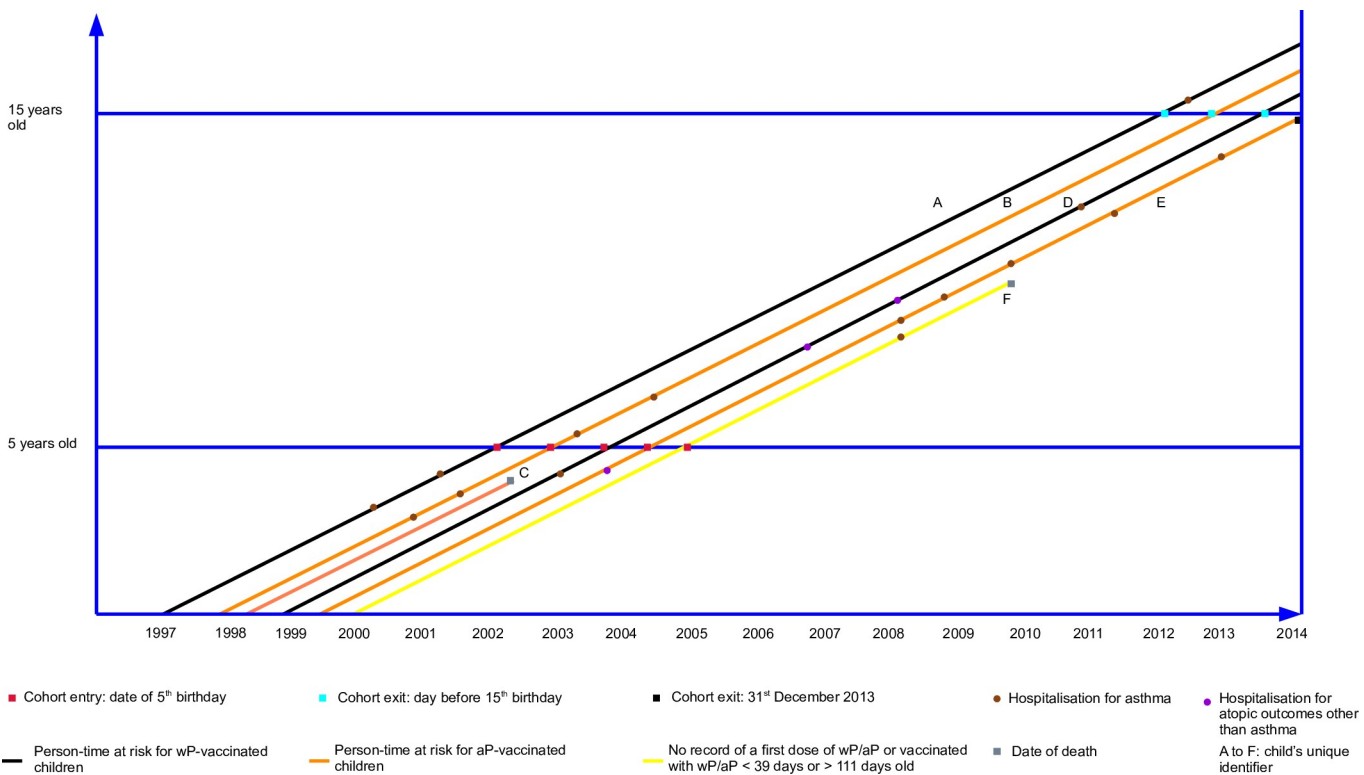

**Fig 1. Lexis diagram (before data cleaning).** Legend: Person-time-at-risk experience of children born in WA or NSW between 1 January 1997 and 31 December 1999 before data cleaning.

## Statistical methods

**Hospitalisations for asthma.** We will compare rates of hospital admissions for asthma in children aged between 5 and 15 years old by means of unadjusted and adjusted Cox regression models for the time-to-first event [31], and Andersen-Gill models for recurrent events using robust standard errors to account for within-child dependence [32].

The person-time-at-risk is depicted in the following Lexis diagrams (Figs 1 and 2).

Each child will enter the cohort on the date of their fifth birthday and the analysis time will be based on days since entry to the cohort. For time-to-first event analyses, censoring will occur at the earliest of the following: first admission to hospital ICD-coded as 'asthma,' the date of death, the day before the 15th birthday, or the end of follow-up (31 December 2013). The same rules will be applied to time-to-recurrent event analyses except there will not be censoring at the time of the first hospitalisation for the outcome of interest. Multivariable models will be stratified by jurisdiction of birth and, where required, non-linear elements (p-splines) will be incorporated.

**Hospitalisations for other atopic outcomes.** We will compare rates of admission for all-cause anaphylaxis, food anaphylaxis, venom, urticaria and atopic dermatitis by means of unadjusted and adjusted Cox regression models for the time-to-first event [31], and Andersen-Gill models for recurrent events using robust standard errors to account for within-child dependence [32]. For children born in WA, the cohort entry will be set at 4 months old, and vaccination status will be treated as a time-dependent covariate. For children born in NSW, the cohort entry will be set at 5 years old. Censoring rules will be applied as above.

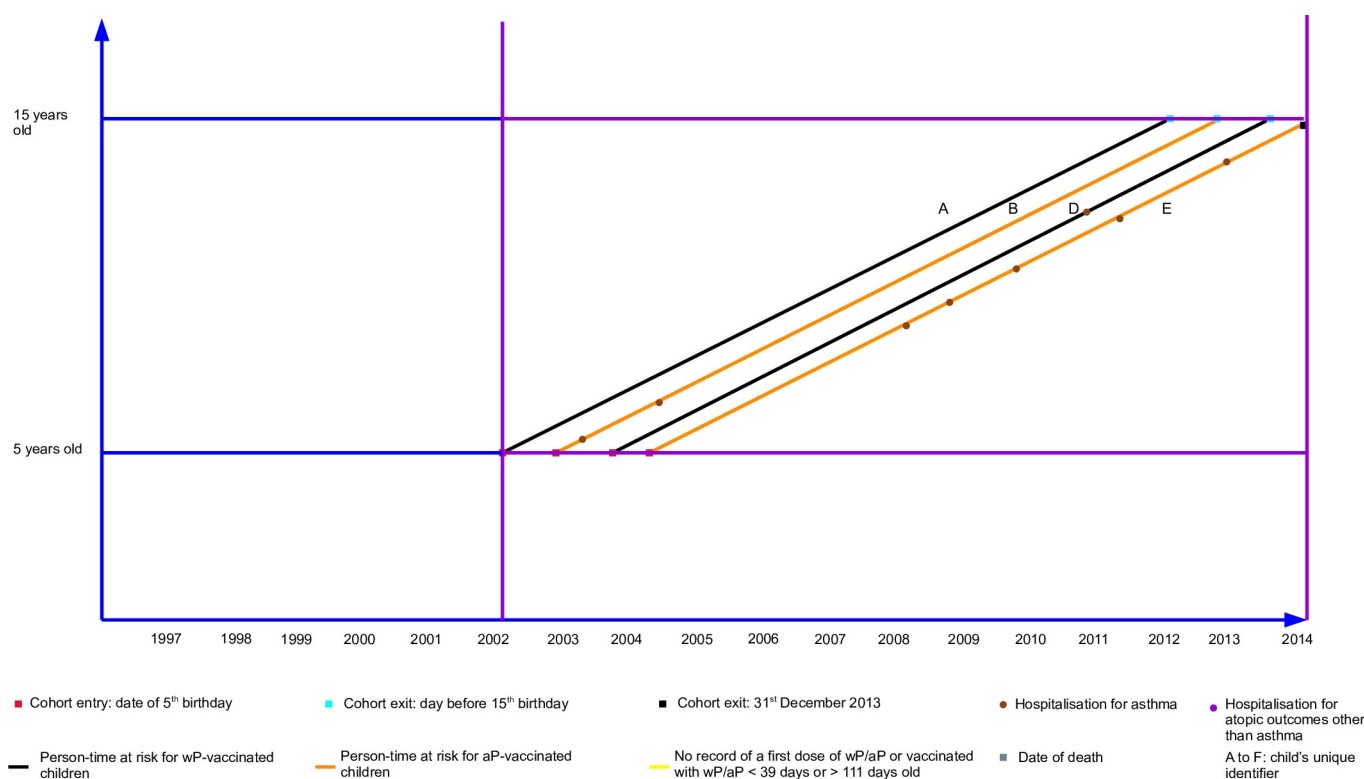

**Fig 2. Lexis diagram (after data cleaning).** Legend: Person-time-at-risk experience of children born in WA or NSW between 1 January 1997 and 31 December 1999 after data cleaning.

**Emergency department presentations.** Emergency department presentations will be analysed using the same statistical approach specified for hospitalisations for asthma, with time zero defined in accordance with the data availability (i.e. the date of the fifth birthday for children born in WA, and the date of the eight birthday for those born in NSW).

All hypothesis tests will be two-sided at 0.05 significance level. All statistical analyses will be carried out in R [33], unless otherwise stated, using the Secured Unified Research Environment from the Sax Institute, NSW.

**Sensitivity analyses.** If numbers allow, we will run sensitivity analyses on the case definition of asthma by restricting the primary diagnosis to 'predominantly allergic asthma' (ICD 10-AM: J45.0) and 'extrinsic asthma' (ICD 9-CM: 493.0–493.1); we will also perform a sensitivity analysis on the case definition of all-cause urticaria, by only including admissions coded as 'allergic urticaria' (ICD 9-CM: 708.0 and ICD-10-AM: L50.0).

According to the extent of missing data, where applicable, sensitivity analyses will be run using suitable imputation techniques.

**Subgroup analysis.** It is uncertain whether delaying the administration of a first dose of pertussis-containing vaccine affects the subsequent risk of developing atopic sensitisation or atopic diseases, with observational studies variously reporting a deleterious [34], beneficial [35–37], and no effects [38]. However, to the best of our knowledge, the relationship between atopic outcomes and age at vaccination has only been tested in recipients of the same type of vaccine, but not in a population vaccinated with a first dose of either, wP or aP. In order to explore possible modification of the effect of the immunisation with wP vs aP by age at vaccination, we will carry out subgroup analyses according to the age of the first dose of wP or aP (before or at 90 days of age, or after 90 days of age).

## Bias

The most likely potential sources of bias are described using the Risk Of Bias In Non-randomised Studies-of Interventions (ROBINS-I) framework [39].

### Pre-exposure domains

**Bias due to confounding.** Calendar date of birth, and state of birth are likely to be associated with one or more of the atopic disease outcomes of interest, either through a direct effect on the disease or through an effect on healthcare utilisation conditional upon disease. They are also known predictors of vaccination with wP versus aP during the transition to aP routine vaccination (Figs 3 and 4). Models will therefore be stratified by state of birth and adjusted for date of birth.

Age is associated with the risk of various atopic diseases, and while it is also associated with vaccination, it should not be associated with receipt of wP versus aP after controlling for date of birth; nonetheless, the Cox survival model intrinsically controls for any effect of age on the risk of outcomes [31].

Statistical models will be adjusted for socioeconomic index, accessibility or remoteness index, Aboriginal ethnicity (identified through a validated multi-stage median algorithm) [41], season of birth and birth order (i.e. the relative position of a child in relation to their siblings). These variables are plausibly associated with the atopic outcomes of interest, and might also be systematically different between children vaccinated with wP versus aP (Figs 3 and 4).

The prevalence of childhood asthma among Aboriginal and/or Torres Strait Islander people (hereafter respectfully referred as Aboriginal) has been measured by a number of surveys

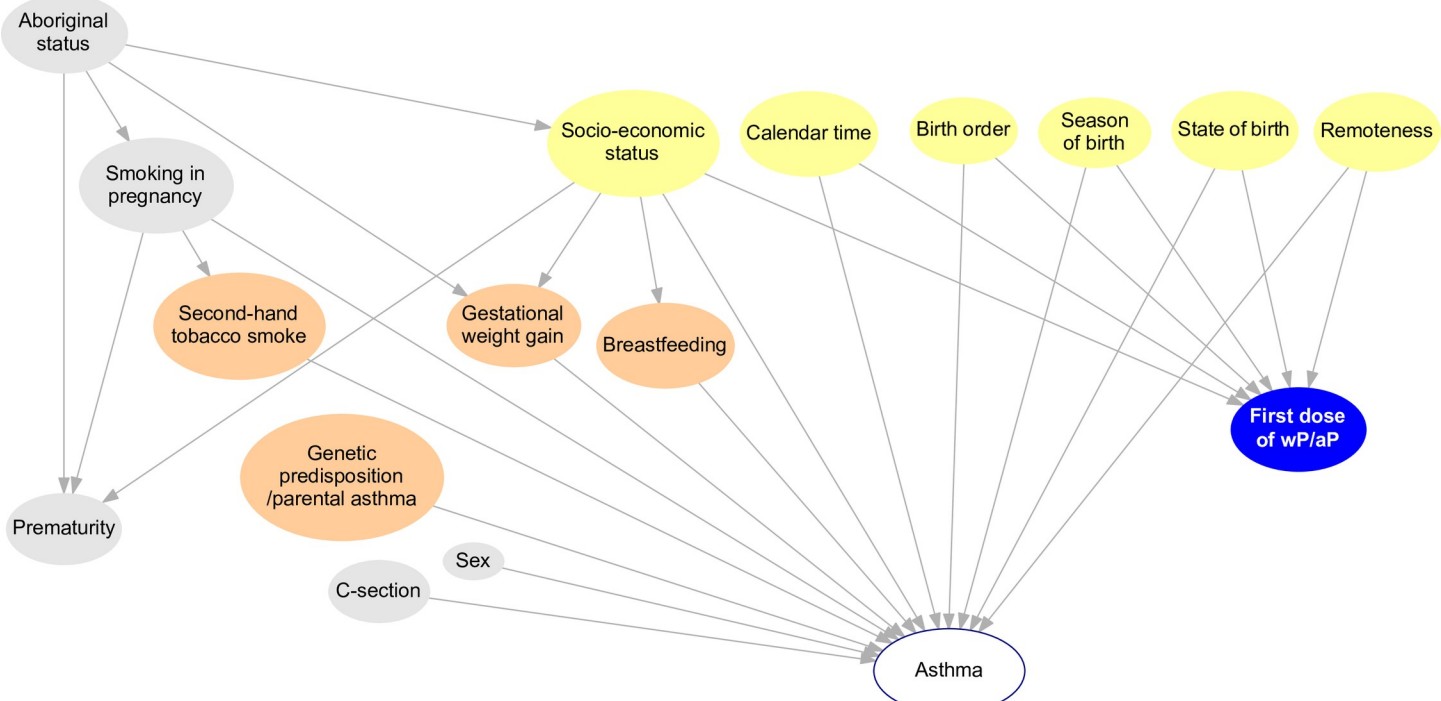

**Fig 3. Direct acyclic graph before adjustment by confounders and baseline covariates.** The blue node represents the exposure (wP or aP); orange and grey nodes: Perinatal, environmental and biological factors at birth/before the exposure that may influence the outcome of interest, but not the receipt of the study vaccines; brown nodes: Confounders of the association between the exposure and atopic asthma; yellow nodes: Adjusted variables; white node: Outcome of interest. Because the association between low birthweight and asthma is mainly driven by gestational age at delivery, birthweight is not depicted in this diagram [40]. C-section: Birth by caesarean. Data are available for all the nodes except for those in orange.

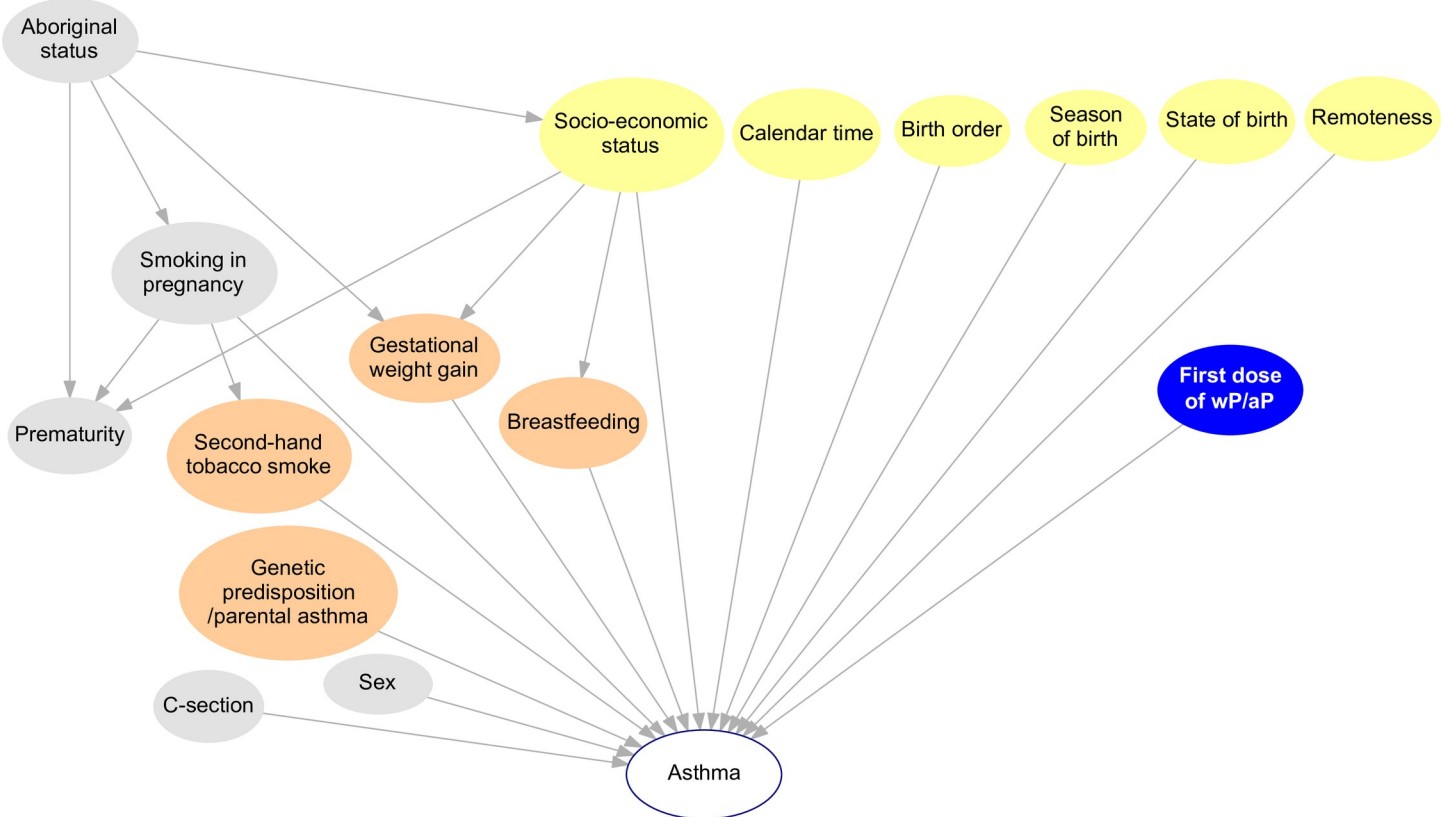

**Fig 4. Direct acyclic graph after adjustment by confounders and baseline covariates.** The blue node represents the exposure (wP or aP); orange and grey nodes: Perinatal, environmental and biological factors at birth/before the exposure that may influence the outcome of interest, but not the receipt of the study vaccines; brown nodes: Confounders of the association between the exposure and atopic asthma; yellow nodes: Adjusted variables; white node: Outcome of interest. Because the association between low birthweight and asthma is mainly driven by gestational age at delivery, birthweight is not depicted in this diagram [40]. C-section: Birth by caesarean. Data are available for all the nodes except for those in orange.

across different settings. According to the most recent National Aboriginal and Torres Strait Islander Health Survey (2018/2019), the prevalence of current and long-term asthma in children under 15 years old was similar for Aboriginal (11.5%; 95% CI 9.6% to 13.4%), and non-Aboriginal children (9.7%; 95% CI 8.5 to 10.9%) [42]. Older studies suggested Aboriginal children living in extremely remote WA may have a lower incidence of asthma across their life course (7.3%; 95% CI 5.3% to 9.7%) than those living in metropolitan Perth (30.5%; 95% CI 27.3% to 33.8%) [43].

Compared to non-Aboriginal Australians, Aboriginal Australians are more likely to experience chronic poverty during their lifespan, intergenerational trauma, smoking during pregnancy, and poor birth outcomes such as prematurity and low-birth-weight newborns [43]. Therefore, any difference in risk of asthma among Aboriginal and non-Aboriginal children may be mediated through differences in socioeconomic status. In the proposed analysis, we intend to adjust for both socioeconomic and Aboriginal status.

The linked datasets do not contain information on some potentially important predictors of asthma or other atopic diseases, including parental and family history, second-hand smoke exposure, maternal obesity, gestational weight gain, or breastfeeding status. However, insofar as these variables are very unlikely to predict the vaccine type received (i.e. they are not likely to be confounders), failure to adjust for them is not expected to bias the effect estimates.

Maternal smoking during pregnancy is captured in the perinatal datasets, and is likely to be a proxy for second-hand smoke exposure. Adjustment for this prognostically important covariate may improve the precision of our effect estimates.

The *sibling effect* refers to the protection against the development of atopic diseases conferred by having siblings. This concept was described for the first time in a cross-sectional analysis from the British Birth Survey in 1986, as cited by Karmaus et al [44], and thought to be related to the hygiene hypothesis proposed by Strachan in 1989 [45].

The representations of the sibling effect in the literature vary widely and include birth order, the size of the sibship, or the number of brothers [46]. Although this phenomenon is supported by a number of epidemiologic studies, its underpinning mechanisms are not well understood. Briefly, increased birth order has been proposed as a proxy of frequent exposure to microorganisms, which are transmitted through unhygienic contact with older siblings during early childhood [45]. These include airway pathogens with $Th_1$ and $Th_{17}$ polarising properties, that are speculated to shift the infant's $Th_2$-biased *in utero* immunophenotype, towards allergy protective immune responses [47]. However, the sibling effect may also be a consequence of *in utero* immune programming from previous pregnancies. For example, increased birth order has been associated with lower levels of IgE in cord blood, and possibly with a lower risk of atopic sensitisation [44].

During the period of transition to aP-only doses, the receipt of wP or aP as a first dose of the primary series should have been driven by their availability in primary care centres only. Nonetheless, it is reasonable that parents would have requested the preferential administration of aP vaccine if a previous child (sibling) experienced any adverse reaction following vaccination with wP. Due to the possibility of confounding, we will include the number of previous pregnancies (a surrogate of birth order) as a variable in the models.

In addition, residual confounding that may lead to biased estimates will be approached using injury, trauma and poisoning-related ICD codes, as negative control outcomes (Table 1).

**Bias in selection of participants into the study (collider bias conditioning enrolment on survival).**    Collider bias arises from conditioning on a common effect of the exposure and the outcome [48]. This type of bias can emerge through the process of selecting individuals into the analysis (i.e. selection bias) [48].

There are two potential sources of bias by delaying cohort entry, or failure to synchronise the start of follow-up with the time of exposure (i.e. first dose of pertussis-containing vaccine).

Firstly, if there exists a causal effect which is different among younger and older children, any effect measured in older children will be systematically different from the overall effect. Secondly, bias could theoretically occur if vaccination of wP versus aP impacts on the likelihood of children surviving to cohort entry. For example if wP vaccine is less effective than aP for preventing death from pertussis disease. In the absence of any causal relationship between pertussis vaccine and atopic disease, the cohort of survivors at 5 years old will be relatively depleted of wP-vaccinated children (who are more likely to have died from pertussis) and those with atopic disease (who are more likely to have died from anaphylaxis) before cohort entry; children who are both wP-vaccinated and who also have atopic disease will be especially unlikely to survive to cohort entry. As a result of this depletion, atopic diseases will be less common among wP vaccinated than aP vaccinated children who survive to cohort entry, even though wP has no protective effect. Nonetheless, in our context, infant and child mortality rates are very low and deaths from pertussis or atopic diseases are rare, so any collider bias is presumably negligible. Assuming no residual confounding, effect estimates are likely to be accurate from the chosen time zero until the end of follow-up. The same explanation applies to conditioning enrolment on survival at 8 years old.

### At-exposure domain

**Bias in classification of the exposure.**   Although the accuracy of the AIR is difficult to validate, by 2001 the underestimation of the immunisation coverage was between 2.7 and 5% [49], suggesting that some vaccine doses were not ascertained.

While it is plausible that some doses of aP were mis-recorded as wP (and vice versa), there is no reason to expect that errors in classification of the vaccine received are differential with respect to ascertainment of the outcomes of interest, since the latter occurred after the exposure, and were recorded through independent mechanisms. As such, any misclassification of vaccination status is likely to bias toward a null effect.

### Post-exposure domains

**Bias due to missing data (selection bias).**   The available administrative data do not identify whether a child emigrated from the state of birth. Therefore, hospital encounters that occurred outside of NSW and WA will not be ascertained. For those children who migrated overseas, data may also be missing for one or more doses of pertussis-containing vaccine. However, as cited by Gidding et al [22], in 2011 just 1.2% of the residents of NSW and WA migrated interstate, and 0.8% migrated overseas. In either case, the rates of interstate migration or emigration from Australia are unlikely to be different for wP versus aP vaccinated children.

According to the 2016 Australian Census, 34.5% and 39.7% of the population of NSW and WA was born overseas, respectively [50,51]. Because the chosen cohort only includes children born in NSW or WA, findings from this study might not be generalisable to immigrant children living in these jurisdictions [22].

**Bias in measurement of the outcome (information bias).**   Concerns regarding the changes of anaphylaxis-related hospitalisation codes, their impact on epidemiologic trends, and further incomplete identification of anaphylaxis cases have been described everywhere else [52], Our study will only ascertain hospitalisations or emergency department presentations where severe IgE-mediated hypersensitivity is implied (i.e. anaphylactic shock) or those labelled as anaphylactic reaction. We expect our approach will be specific for IgE-mediated disease outcomes, but at the expense of sensitivity. Therefore, the proposed analysis plan is unlikely to fully capture the burden of IgE-mediated atopic disease morbidity since many mild cases would not have been hospitalised, or may have received alternative codes [13].

Furthermore, random errors in the identification of the outcome might be possible since ICD codes were used to determine hospitalisations for asthma and other atopic diseases. Any misclassification of the outcome will be non-differential and likely to bias toward the null. Likewise, presentations to the emergency department for the outcomes of interest may be miscoded, as diagnoses are primarily ascribed by clinical and clerical personnel at the emergency facilities who are not specialised in clinical coding.

## Discussion

The proposed observational design will use administrative data to assess the hazard of hospital admission and emergency department presentation for allergic diseases in a birth cohort of children born in Australia during the transition from using wP to aP priming vaccine doses.

The limitations of this study have been described in the "Bias" section of this manuscript. Although more sophisticated techniques have been proposed in the causal inference literature for the selection of covariates and further statistical modelling [27,53], features inherent to the transition period, in particular the apparently untargeted manner in which wP versus aP vaccine doses were administered, may still enable us to treat these data as though from a

randomised experiment after controlling for the two predictors of wP versus aP vaccination, namely the date and state of birth.

The aim of the proposed analysis plan is to assess for a possible causal effect of wP on prevention of asthma and other atopic diseases, although the potential for residual confounding and other biases means any protective association is unlikely to be sufficient to change policy. If support is provided, the results of this study could motivate and inform the implementation of randomised controlled trials in Australia, where wP has been phased out from the primary series of pertussis immunisation, and where the burden of atopic diseases in childhood is high [54].

## Ethics and dissemination of results

This study has been approved by the human research ethics committees of the Department of Health of WA (approval ID: 2012/75), NSW Population Health Service (HREC/13/CIPHS/15), Australian Institute of Health and Welfare (EC 2012/4/62), Curtin University (HRE2019-0350) as well as by the WA Aboriginal Health Ethics Committee (approval ID: 459), and the Aboriginal Health and Medical Research Council Ethics Committee (approval ID: 931/13).

Research findings will be disseminated through peer-reviewed journals, conference presentations, plain language summaries and electronic and social media platforms.

We would also like to thank Dr Yue Wu (Health and Clinical Analytics Lab, Sydney School of Public Health, Faculty of Medicine and Health) for her critical input on the development and further edits of the DAGs.

## Acknowledgments

We thank the staff at the Population Health Research Network (PHRN) data linkage and infrastructure nodes (the WA Data Linkage Branch, the NSW Centre for Health Record Linkage, and the Australian Institute for Health and Welfare), the WA and Commonwealth Departments of Health and NSW Ministry of Health who provided advice and the data, and the Wesfarmers Centre of Vaccines and Infectious Diseases Community Reference Group for their valuable insights. Our data sources are acknowledged below:

1. Perinatal data: NSW Perinatal Data Collection and WA Midwives Notification System.

2. Births: NSW Birth Registration Data Collection and WA Registry of Births Deaths and Marriage.

3. Death data: National Death Index.

4. Immunisation data: the AIR dataset.

5. Hospitalisation data: NSW Admitted Patient Data Collection and WA Hospital Morbidity Data Collection.

6. Emergency department data: Emergency Department Data Collection (NSW and WA).

## Author Contributions

**Conceptualization:** Gladymar Pérez Chacón, Peter C. Richmond, Thomas L. Snelling.

**Funding acquisition:** Heather F. Gidding, Hannah C. Moore.

**Methodology:** Gladymar Pérez Chacón, Mark Jones, Thomas L. Snelling.

**Supervision:** Peter C. Richmond, Heather F. Gidding, Hannah C. Moore, Thomas L. Snelling.

**Writing – original draft:** Gladymar Pérez Chacón.

**Writing – review & editing:** Parveen Fathima, Mark Jones, Rosanne Barnes, Peter C. Richmond, Heather F. Gidding, Hannah C. Moore, Thomas L. Snelling.

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
