## [Decision Letter · Decision Letter 0]

4 Oct 2021

PONE-D-21-25854Pertussis immunisation in infancy and atopic outcomes: A protocol for a population-based cohort study using linked administrative dataPLOS ONE

Dear Dr. Pérez Chacón1,

Thank you for submitting your manuscript to PLOS ONE. After careful consideration, we feel that it has merit but does not fully meet PLOS ONE’s publication criteria as it currently stands. Therefore, we invite you to submit a revised version of the manuscript that addresses the points raised during the review process.

ACADEMIC EDITOR: Please insert comments here and delete this placeholder text when finished. Be sure to:Indicate which changes you require for acceptance versus which changes you recommendAddress any conflicts between the reviews so that it's clear which advice the authors should followProvide specific feedback from your evaluation of the manuscriptPlease ensure that your decision is justified on PLOS ONE’s publication criteria and not, for example, on novelty or perceived impact.

We look forward to receiving your revised manuscript.

Kind regards,

Lucinda Shen, 

Staff Editor

on behalf of 

Daniela Flavia Hozbor

Academic Editor 

PLOS ONE

Journal Requirements:

Additional Editor Comments (if provided):

Reviewers' comments:

Reviewer's Responses to Questions

**Comments to the Author**

1. Does the manuscript provide a valid rationale for the proposed study, with clearly identified and justified research questions?

Reviewer #1: Yes

Reviewer #2: Yes

2. Is the protocol technically sound and planned in a manner that will lead to a meaningful outcome and allow testing the stated hypotheses?

Reviewer #1: Yes

Reviewer #2: Yes

3. Is the methodology feasible and described in sufficient detail to allow the work to be replicable?

Reviewer #1: Yes

Reviewer #2: Yes

4. Have the authors described where all data underlying the findings will be made available when the study is complete?

Reviewer #1: Yes

Reviewer #2: Yes

5. Is the manuscript presented in an intelligible fashion and written in standard English?

Reviewer #1: Yes

Reviewer #2: Yes

6. Review Comments to the Author

You may also provide optional suggestions and comments to authors that they might find helpful in planning their study.

Reviewer #1: This seems a straightforward retrospective cohort study that has the potential to answer an important question. I like the design and would like to commend the authors for considering this study. Its use of retrospective data is a major advantage and will hopefully get to useful answers quickly and relatively cheaply. I do have a few minor questions and comments for the authors to consider:

Lines 55-56: Although this is not core to the protocol, it is important to note that Yeung’s modelling study actually indicates that significant inequities persist with the poorest countries carrying the brunt of both pertussis morbidity and death. The EPI as a framework has been great but deal less with capacity that increases access to vaccines although other access programs were later leveraged on it (e.g. GAVI).

Line 65: ‘more or less stochastic’ – I did smile a bit at the wording because being a disciple of C.S Lewis I would have gone for the simple ‘random’, in part because statisticians tend to technically use ‘stochastic’ to technically refer to deliberately created randomness.

Lines 80-82: I think it is important to note that a decrease in asthma hospitalisations should not be read to imply a decrease in asthma as there has been a significant improvement over time in managing asthma in ways that reduce severe exacerbations. What would be useful would be to know the prevalence over the period. Still I understand why this is mentioned here given the anticipated outcome.

Lines 89: Is that food-induced or just anaphylaxis in general?

Line 105: Did the period of shortage eliminate wP from the schedule or did it just reduce the number of wP doses in the schedule?

Line 113: ‘primed with wP’: It may be important to note that in ‘pertussis cycles’, the word ‘prime’ has been used both to refer to the first dose of the vaccine given (a child who gets a single dose of wP as first dose but received aP for the balance of the schedule would thus have been wP primed) as well as the completing of a 3-dose primary schedule. This may be important here as the study may need to take this distinction into consideration in the context of ‘mixed’ vaccination.

Lines 168-172: Will this require a need to consider stratifying by or adjusting for aP type? wP seems simple enough as there was only one manufacturer over the period under review. Pertussis vaccines, even if same ‘type’ differ with respect to how they are manufactured, adjuvants used, constituted, etc., All these have an in impact on immunogenicity and efficacy.

Line 199: ‘hospitalisations for asthma before 5 years old will be disregarded’ : I am reading this to mean that they will not count as an outcome rather than that they will be ignored. Age at onset of asthma, and number of previous hospitalisations have been known to be important predictors of future need for hospitalisation. So, am I hoping that the authors intend to note this if as a minimum to consider it as a potential confounder that may require some adjusting for. Having said that, and having read the reasons for excluding events before 5 years of age, I cannot figure out why disregarding this period is sensible if the data were available especially given the introduction section that shows an increase in allergic reaction in younger children. Why would it not be more reasonable to rather have to rather stratify the outcomes by the ages of the cohort? As such I would assume that the second reason given, that of lack of data before 2001 would be the only defensible reason in my opinion.

Table 2. ‘Index of Relative Socio-economic Advantage and Disadvantage’: Will that be at the time of vaccination, start of time under evaluation (5 years) or at the time of outcome? Or does it not matter?

Lines 275-284 + Figures 2. This may be just my understanding of the design that leads to this comment on the descriptions of this Lexis diagram. In this design, a cohort would be defined by exposure and that exposure occurs in infancy, quite close to birth. Does that then not mean that each child enters the cohort at the verifiable time of exposure, even if assessed time at risk of outcome is only entered after 5 years? Or does the lack of data alluded to before affect this understanding?

Finally, while I do appreciate the fact that the authors have spent some time on bias as the major concern, I would have liked to see a distinct reflective section on ‘Limitations’ to acknowledge other potential limitations, even if briefly. Are there not other limitations? What about confounding for example. I absolutely like the fact that the authors have chosen to use DAGs to decide on potential factors, but even the chosen model is determined and limited by causal assumptions made. For example, some may argue that admissions before 5 years of age as predictor of further admissions are missing from the model; should genetic predisposition and parental asthma be two independent variables, etc.

Reviewer #2: The burden of IgE-mediated food allergy in Australian born children is reported to be among the highest globally. This illness shares risk factors and frequently coexists with asthma, one of the most common noncommunicable diseases of childhood. Findings from a case-control study suggest that compared to immunisation with acellular pertussis vaccine, early priming of infants with whole-cell pertussis vaccine may be associated with a lower risk of subsequent IgE-mediated food allergy.

The authors propose a protocol for a population-based cohort study using linked administrative. The observational design will use administrative data to assess the hazard of hospital admission and emergency department presentation for allergic diseases in a birth cohort of children born in Australia during the transition from using wP to aP priming vaccine doses.

The authors describe the methods to compare a birth cohort of children vaccinated with a first dose of wP versus aP with respect to the time to a first or recurrent tertiary care encounter for acute exacerbations of asthma and other atopic diseases. The strengths and limitations of this study are clearly identified and described in detail.

I suggest that authors mention how the results obtained in this study would be complemented with those obtained from the OPTIMUM study (Perez Chacon G, Estcourt MJ, Totterdell J, et al. OPTIMUM study protocol: an adaptive randomised controlled trial of a mixed whole-cell/acellular pertussis vaccine schedule. BMJ Open 2020;10:e042838. doi:10.1136/ bmjopen-2020-042838)

7. PLOS authors have the option to publish the peer review history of their article (what does this mean?). If published, this will include your full peer review and any attached files.

Reviewer #1: **Yes: **Rudzani Muloiwa

Reviewer #2: No

---

## [Author Response · Author response to Decision Letter 0]

15 Oct 2021

15 October 2021

Dear Lucinda Shen and Dr Daniela Flavia Hozbor

Re PONE-D-21-25854

Thank you for considering our manuscript entitled 

“Pertussis immunisation in infancy and atopic outcomes: 

A protocol for a population-based cohort study using linked 

administrative data” for publication in PLOS One under the registered report scheme. Below we attach a response letter to our reviewers’ comments. In addition, the revised version includes changes in the reference list, and minor edits to the figures and tables. These have all been acknowledged in tracked change copy of and in the following rebuttal.

Reviewer #1

This seems a straightforward retrospective cohort study that has the potential to answer an important question. I like the design and would like to commend the authors for considering this study. Its use of retrospective data is a major advantage and will hopefully get to useful answers quickly and relatively cheaply. I do have a few minor questions and comments for the authors to consider:

Comment 1. Lines 55-56: Although this is not core to the protocol, it is important to note that Yeung’s modelling study actually indicates that significant inequities persist with the poorest countries carrying the brunt of both pertussis morbidity and death. The EPI as a framework has been great but deal less with capacity that increases access to vaccines although other access programs were later leveraged on it (e.g. GAVI). 

Response 1. We thank Prof Muloiwa for this comment. The following changes have been made to this section of the manuscript: “From 1974, improved access to wP formulations through the Expanded Programme of Immunization has decreased vaccination inequalities in the developing world and the global burden of pertussis disease in children younger than 5 years old.[5] Nevertheless, geographical, social factors and weak health systems still drive vaccine inequity within low and middle income countries, substantially affecting the coverage of the first and third dose of pertussis vaccine primary series, and pertussis-related deaths.[5,6]”

Comment 2. Line 65: ‘more or less stochastic’ – I did smile a bit at the wording because being a disciple of C.S Lewis I would have gone for the simple ‘random’, in part because statisticians tend to technically use ‘stochastic’ to technically refer to deliberately created randomness. 

Response 2. We thank Prof Muloiwa for this comment. Our preference is to keep the wording used in line 65 as it is.

Comment 3. Lines 80-82: I think it is important to note that a decrease in asthma hospitalisations should not be read to imply a decrease in asthma as there has been a significant improvement over time in managing asthma in ways that reduce severe exacerbations. What would be useful would be to know the prevalence over the period. Still I understand why this is mentioned here given the anticipated outcome.

Response 3. We thank Prof Muloiwa for this comment. This section has been rewritten and it now reads: “In Australia the prevalence of self-reported “current asthma” in children under 15 years old has declined from 13.5% in 2001 to 9.9% in 2007/2008.[11] While the rates of all-cause hospitalisation remained stable, paediatric admissions to hospital coded as asthma decreased nationwide by 33% from 1998/1999 to 2010/2011.[12] Optimisation in the management of acute exacerbations by carers, modifications in admission practices or in the severity of the presentations have been proposed as potential explanations for this trend, yet it remains uncertain the role played by these factors over this period.[12] Admissions to hospital are more likely to occur in those with more severe symptoms, or in those who did not respond well to the initial management in the emergency department. On the other hand, visits to the emergency not only reflect disease severity, but in some circumstances, socio-demographic disparities that negatively impact the access to primary care.”

Comment 4. Lines 89: Is that food-induced or just anaphylaxis in general?

Response 4. Food anaphylaxis

Comment 5. Line 105: Did the period of shortage eliminate wP from the schedule or did it just reduce the number of wP doses in the schedule?

Response 5. Authors’ reply: We thank Prof Muloiwa for this question. Clarification has been provided regarding the period of shortage of wP in the Isle of Wight (Venter, 2016). This section now reads: “Similarly, a retrospective cohort study of children born in the Isle of Wight (UK) between September 2001 and August 2002 (a period of shortage of wP) did not show an association either between the type of pertussis-containing vaccine received as a first dose, or the type of pertussis immunization schedule (i.e. wP-only doses versus at least one dose of aP in fully-vaccinated infants for pertussis antigens), and IgE-mediated food allergy, atopic dermatitis and asthma during a 10-year period of follow-up.[17]”

 Comment 6. Line 113: ‘primed with wP’: It may be important to note that in ‘pertussis cycles’, the word ‘prime’ has been used both to refer to the first dose of the vaccine given (a child who gets a single dose of wP as first dose but received aP for the balance of the schedule would thus have been wP primed) as well as the completing of a 3-dose primary schedule. This may be important here as the study may need to take this distinction into consideration in the context of ‘mixed’ vaccination.

Response 6. Authors’ reply: We thank Prof Muloiwa for this comment. In this case, priming refers to the first dose of pertussis immunisation schedule. This section now reads: “Conversely, a case-control study of Australian infants born between 1997 and 1999, the period of changeover to aP-based schedules, reported a lower likelihood of IgE-mediated food allergy in infants primed with a first dose of wP, compared to those that received a first dose of aP [odds ratio: 0.77; 95% confidence interval (CI), 0.62 to 0.95].[19]”

 Comment 7. Lines 168-172: Will this require a need to consider stratifying by or adjusting for aP type? wP seems simple enough as there was only one manufacturer over the period under review. Pertussis vaccines, even if same ‘type’ differ with respect to how they are manufactured, adjuvants used, constituted, etc., All these have an in impact on immunogenicity and efficacy.

Response 7. Authors’ reply: We thank Prof Muloiwa for this suggestion, unfortunately, we do not have information regarding the brands of pertussis-containing vaccines/vaccine manufacturers in the dataset provided for this project.

Comment 8. Line 199: ‘hospitalisations for asthma before 5 years old will be disregarded’: I am reading this to mean that they will not count as an outcome rather than that they will be ignored. Age at onset of asthma, and number of previous hospitalisations have been known to be important predictors of future need for hospitalisation. So, am I hoping that the authors intend to note this if as a minimum to consider it as a potential confounder that may require some adjusting for. Having said that, and having read the reasons for excluding events before 5 years of age, I cannot figure out why disregarding this period is sensible if the data were available especially given the introduction section that shows an increase in allergic reaction in younger children. Why would it not be more reasonable to rather have to rather stratify the outcomes by the ages of the cohort? As such I would assume that the second reason given, that of lack of data before 2001 would be the only defensible reason in my opinion.

Response 8. We thank Prof Muloiwa for this comment. This revised version of the manuscript clarifies the principles that govern the selection of covariates and the reasons why post exposure variables (i.e. hospitalisations for asthma before 5 years old) will not be considered for regression. These are listed in the section “Confounders and explanatory variables”, which now reads: “Maternal and infant characteristics as recorded on the birth registries and perinatal data collections, will be incorporated in the models as potential confounders of vaccine effects on atopic disease (i.e. as common causes of the exposure and the outcomes of interest), or as prognostically important covariates. Both will be selected a priori based on causal considerations and the modified disjunctive cause criterion proposed by VanderWeele,[27] aided by graphical models of asthma generated in GeNIe modeler,[28] and other atopic diseases (see Estcourt et al [19]). Briefly, these will be restricted to:

• common causes of the exposure and outcome;

• the proxies of unmeasured confounders; and

• those related to the outcome, but not to the exposure.

The proposed set was measured before the administration of the first dose of wP/aP. Collider stratification bias will be avoided by not including variables in the causal pathway from the exposure to the outcome (i.e. those that occur post exposure).”

Comment 9. Table 2. ‘Index of Relative Socio-economic Advantage and Disadvantage’: Will that be at the time of vaccination, start of time under evaluation (5 years) or at the time of outcome? Or does it not matter?

Response 9. We thank Prof Muloiwa for this question/comment. In the original version of the manuscript, we specified that the Index of Relative Socio-economic Advantage and Disadvantage is a national measure of geographic remoteness and access to services for populated localities throughout Australia. This is determined according to the residential address of the mother at the time of their child’s birth. The considerations detailed in response 8, are also applicable to comment 9.

Comment 10. Lines 275-284 + Figures 2. This may be just my understanding of the design that leads to this comment on the descriptions of this Lexis diagram. In this design, a cohort would be defined by exposure and that exposure occurs in infancy, quite close to birth. Does that then not mean that each child enters the cohort at the verifiable time of exposure, even if assessed time at risk of outcome is only entered after 5 years? Or does the lack of data alluded to before affect this understanding?

Response 10. We thank Prof Muloiwa for these questions/comments. Although children received the exposure close to birth, the cohort entry is delayed until the age of five years old (for hospital admissions ICD-coded as asthma, and admissions to hospital for other atopic outcomes in children born in NSW). We argue that collider bias is not a major concern because in the study settings, deaths for atopic diseases or pertussis are negligible.

Comment 11. Finally, while I do appreciate the fact that the authors have spent some time on bias as the major concern, I would have liked to see a distinct reflective section on ‘Limitations’ to acknowledge other potential limitations, even if briefly. Are there not other limitations? What about confounding for example. I absolutely like the fact that the authors have chosen to use DAGs to decide on potential factors, but even the chosen model is determined and limited by causal assumptions made. For example, some may argue that admissions before 5 years of age as predictor of further admissions are missing from the model; should genetic predisposition and parental asthma be two independent variables, etc.

Response 11. We thank Prof Muloiwa for raising these concerns. 

The DAGs have been edited accordingly. 

• Genetic predisposition and parental asthma are included in a single node. 

• Season of birth has now been added as confounder and will also be summarised in the description of the cohort (this has also been reflected on Table 2). 

• In addition, the biasing paths were deleted from Figure 4 (DAG after adjustment for confounders) and to ensure readability, we changed the placement of the nodes.

• For the reasons described in comment 8, we will not include previous admissions to hospital for asthma in the models. These, however, will be summarised in the final report using descriptive statistics.

Although the limitations of the study have been thoroughly assessed throughout the “Bias” section. Nevertheless, the following has been added to the discussion section, which now reads: The limitations of this study have been described in the “Bias” section of this manuscript. Although more sophisticated techniques have been proposed in the causal inference literature for the selection of covariates and further statistical modelling,[27,53] features inherent to the transition period, in particular the apparently untargeted manner in which wP versus aP vaccine doses were administered, may still enable us to treat these data as though from a randomised experiment after controlling for the two predictors of wP versus aP vaccination, namely the date and state of birth.

Reviewer #2: 

Comment 1. The burden of IgE-mediated food allergy in Australian born children is reported to be among the highest globally. This illness shares risk factors and frequently coexists with asthma, one of the most common noncommunicable diseases of childhood. Findings from a case-control study suggest that compared to immunisation with acellular pertussis vaccine, early priming of infants with whole-cell pertussis vaccine may be associated with a lower risk of subsequent IgE-mediated food allergy.

The authors propose a protocol for a population-based cohort study using linked administrative. The observational design will use administrative data to assess the hazard of hospital admission and emergency department presentation for allergic diseases in a birth cohort of children born in Australia during the transition from using wP to aP priming vaccine doses.

The authors describe the methods to compare a birth cohort of children vaccinated with a first dose of wP versus aP with respect to the time to a first or recurrent tertiary care encounter for acute exacerbations of asthma and other atopic diseases. The strengths and limitations of this study are clearly identified and described in detail.

I suggest that authors mention how the results obtained in this study would be complemented with those obtained from the OPTIMUM study (Perez Chacon G, Estcourt MJ, Totterdell J, et al. OPTIMUM study protocol: an adaptive randomised controlled trial of a mixed whole-cell/acellular pertussis vaccine schedule. BMJ Open 2020;10:e042838. doi:10.1136/ bmjopen-2020-042838)

Response 1. We thank the reviewer for this comment. A reference was added to this section, in order to link the proposed observational design with the ongoing OPTIMUM study: “If support is provided, the results of this study could motivate and inform the implementation of randomised controlled trials in Australia, where wP has been phased out from the primary series of pertussis immunisation, and where the burden of atopic diseases in childhood is high.[54]”

Additional minor revisions:

• Table 1. Transposition errors were noted on the ICD 9 and ICD 10 AM combined injury, trauma and poisoning codes. These have been amended. 

• Table 2. The header of this table has been edited.

• Figures 1 and 2. We improved the resolution.

Should you have any questions or require additional clarification, please do not hesitate to contact me.

Sincerely yours,

Gladymar Pérez Chacón

MD&DTM&H,

on behalf of the authors

gladymar.perez@telethonkids.org.au

---

## [Decision Letter · Decision Letter 1]

9 Nov 2021

Pertussis immunisation in infancy and atopic outcomes: A protocol for a population-based cohort study using linked administrative data

PONE-D-21-25854R1

Dear Dr. Gladymar Pérez Chacón,

We’re pleased to inform you that your manuscript has been judged scientifically suitable for publication and will be formally accepted for publication once it meets all outstanding technical requirements.

Kind regards,

Daniela Flavia Hozbor

Academic Editor

PLOS ONE

Additional Editor Comments (optional):

Reviewers' comments:

Reviewer's Responses to Questions

**Comments to the Author**

1. Does the manuscript provide a valid rationale for the proposed study, with clearly identified and justified research questions?

Reviewer #1: Yes

2. Is the protocol technically sound and planned in a manner that will lead to a meaningful outcome and allow testing the stated hypotheses?

Reviewer #1: Yes

3. Is the methodology feasible and described in sufficient detail to allow the work to be replicable?

Reviewer #1: Yes

4. Have the authors described where all data underlying the findings will be made available when the study is complete?

Reviewer #1: Yes

5. Is the manuscript presented in an intelligible fashion and written in standard English?

Reviewer #1: Yes

6. Review Comments to the Author

You may also provide optional suggestions and comments to authors that they might find helpful in planning their study.

Reviewer #1: Thank you for meaningfully engaging with my comments - all really made as suggestions to the authors to improve what I had already indicated was a solid manuscript. I have no further comments to make.

7. PLOS authors have the option to publish the peer review history of their article (what does this mean?). If published, this will include your full peer review and any attached files.

Reviewer #1: **Yes: **Rudzani Muloiwa

---

## [Editor Report · Acceptance letter]

11 Nov 2021

PONE-D-21-25854R1 

Pertussis immunisation in infancy and atopic outcomes: A protocol for a population-based cohort study using linked administrative data 

Dear Dr. Pérez Chacón:

I'm pleased to inform you that your manuscript has been deemed suitable for publication in PLOS ONE. Congratulations! Your manuscript is now with our production department. 

Kind regards, 

on behalf of

Dr. Daniela Flavia Hozbor 

Academic Editor

PLOS ONE